# LeCO-NeRF: Learning Compact Occupancy for Large-scale Neural Radiance Fields

## Abstract

Neural Radiance Fields (NeRFs) have shown impressive results in modeling large-scale scenes. A critical problem is how to effectively estimate the occupancy to guide empty-space skipping and point sampling. Although grid-based methods show their advantages in occupancy estimation for small-scale scenes, large-scale scenes typically have irregular scene bounds and more complex scene geometry and appearance distributions, which present severe challenges to the grid-based methods for handling large scenes, because of the limitations of predefined bounding boxes and grid resolutions, and high memory usage for grid updating. In this paper, we propose to learn a compact and efficient occupancy representation of large-scale scenes. Our main contribution is to learn and encode the occupancy of a scene into a compact MLP in an efficient and self-supervised manner. We achieve this by three core designs. *First*, we propose a novel Heterogeneous Mixture of Experts (HMoE) structure with common Scene Experts and a tiny Empty-Space Expert. The heterogeneous structure can be effectively used to model the imbalanced unoccupied and occupied regions in NeRF where unoccupied regions need much fewer parameters. *Second*, we propose a novel imbalanced gate loss for HMoE, motivated by the prior that most of the 3D points are unoccupied. It enables the gating network of HMoE to accurately dispatch the unoccupied and occupied points. *Third*, we also design an explicit density loss to guide the gating network. Then, the occupancy of the entire large-scale scene can be encoded into a very compact gating network of the HMoE. As far as we know, we are the first to learn the compact occupancy of large-scale NeRF by an MLP. We show in the experiments that our occupancy network can very quickly learn more accurate, smooth, and clean occupancy compared to the occupancy grid. With our learned occupancy as guidance for empty space skipping, our method can consistently obtain $2.5\times$ speed-up on the state-of-the-art method Switch-NeRF, while achieving highly competitive performances on several challenging large-scale benchmarks.

## 1 Introduction

Neural Radiance Fields (NeRF) (Mildenhall et al., 2020) have been used to model large-scale 3D scenes by scene decomposition, such as Mega-NeRF (Turki et al., 2022), Block-NeRF(Tancik et al., 2022), and Switch-NeRF (MI & Xu, 2023). Although they have achieved promising performances, the critical problem of modeling occupancy for large-scale scenes remains under-explored. A large 3D scene is usually very sparse, with a large portion of the 3D scene as empty spaces. Thus, modeling the occupancy can effectively guide the empty-space skipping and point sampling. Using an occupancy grid to guide the point sampling has become a common practice in small-scale NeRF (Müller et al., 2022; Fridovich-Keil et al., 2022; Hu et al., 2022; Li et al., 2022). The occupancy grid stores density and occupancy in grid cells. The occupancy is computed from the density value by evaluating the NeRF network and defining a density threshold. During the training of NeRF, they sample 3D points from the grid cells and compute new density values to update the occupancy in a momentum way. The computation of updating the grid is thus related to the resolution of grids.

The occupancy grid works well for modeling the occupancy of small-scale scenes. However, it has clear limitations on large-scale scenes. Firstly, the memory used to store the grid and the computation used to update it increase significantly along with the grid resolution. This limits the grid from increasing its resolution to model large-scale scenes in detail. Secondly, the occupancy grid needs

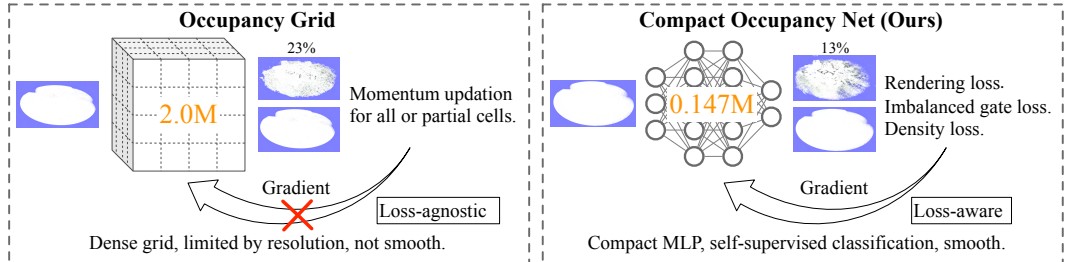

Figure 1: Illustration of the differences between the occupancy grid and our occupancy network. Our occupancy network is a compact MLP with only 0.147M parameters, trained by our designed losses within HMoE. The occupancy grid stores 2.0M parameters with a resolution of $128^3$. It needs is not aware of the training losses. The images are the visualization of the occupied and unoccupied parts as stated in 4.2.

more prior knowledge of the geometry of the scene. The scene should be more regular so that it can be bounded by a tight bounding box. Thirdly, most of the grids are unoccupied due to the sparsity of the scene, making the grid not compact enough and wasting memory and computation. Finally, the momentum updating of the occupancy grid is not directly related to the actual rendering loss, making it agnostic to the actual rendering quality, leading to unsatisfactory results. These limitations make the occupancy grid challenging to be directly applied to model complex large-scale scenes.

To tackle the challenges of modeling occupancy for large-scale scenes, in this paper, we propose LeCO-NeRF to learn a compact occupancy representation with an MLP. Fig. 1 compares the occupancy grid and our occupancy network. An essential nature of a 3D scene is that the occupied points are much fewer than the unoccupied points, while containing significantly more important information. Therefore, modeling occupancy is naturally very imbalanced and heterogeneous. This motivates us to propose a Heterogeneous Mixture of Experts (HMoE) network, an imbalanced gate loss, and a density loss to learn the occupancy. Our contributions are discussed below.

**Firstly**, we propose a novel Heterogeneous Mixture of Experts (HMoE) network to learn the occupancy of a large-scale scene. The HMoE network consists of several Scene Experts designed to encode the occupied points. It contains another special Empty Space Expert designed to handle the unoccupied 3D points. A compact gating network is used by HMoE to determine which expert a 3D point should be dispatched into. If a 3D point is dispatched to the Empty Space Expert, this point is seen as unoccupied. Therefore, the gating network can serve as a representation of the occupancy. The HMoE is heterogeneous in the structure of experts. The empty space expert is designed to be a very *tiny* network with much fewer parameters than the scene experts, as the unoccupied points are less informative and much easier to model. **Secondly**, we propose an imbalanced gate loss for the gating network in HMoE. Since a large portion of the space is unoccupied, the decision of our gating network should explicitly model the imbalance of occupancy. We accordingly design an imbalanced gate loss to make the gating network dispatch a large portion of the 3D points to the empty space expert, instead of dispatching samples uniformly as in previous MoE methods (Lepikhin et al., 2021; Fedus et al., 2022; MI & Xu, 2023). With the two designs of HMoE and imbalanced gate loss, we find that the gating network can already effectively distinguish the occupied and unoccupied points implicitly. **Thirdly**, to better learn the occupancy of a large-scale scene, we further propose a density loss to guide the training of the gating network. In a NeRF representation, the density of an unoccupied point is much smaller than that of an occupied point. We explicitly use this constraint to design a density loss to make the gating network dispatch points with small density values to the Empty Space Expert. This density loss can ensure the network predicts more accurate occupancy.

Our LeCO-NeRF converges very fast in learning the scene occupancy. The imbalanced gate loss and the density loss also work together with the rendering loss, so that our network is more aware of the rendering quality. After training the occupancy, we can utilize it to guide the point sampling in the state-of-the-art large-scale NeRF method, *i.e.* Switch-NeRF (MI & Xu, 2023). We freeze the learned occupancy network and use it as an occupancy predictor. If a point is predicted as unoccupied, it is discarded and is not processed by the main NeRF network. In our experiments, we can consistently outperform the occupancy grid in terms of accuracy, and achieve $2.5\times$ acceleration compared to Switch-NeRF, while obtaining highly competitive performance. Our method can also learn more accurate, smooth, and clean occupancy compared to the occupancy grid. The smoothness is apparent as shown in the rendered videos in the supplementary.

## 2 RELATED WORK

**NeRF.** Neural Radiance Fileds (Mildenhall et al., 2020) utilize a multilayer perceptron (MLP) network to encode a 3D scene from multi-view images. It has been extended to model a lot of tasks Liu et al. (2020); Xu et al. (2022); Kerbl et al. (2023); Zhang et al. (2023) or even city-level large-scale scenes (Turki et al., 2022; Tancik et al., 2022; MI & Xu, 2023). The main idea of these large-scale NeRF methods is to decompose the large-scale scene into partitions and use different sub-networks to encode different parts, and then compose the sub-networks. The Mega-NeRF (Turki et al., 2022) and Block-NeRF (Tancik et al., 2022) manually decompose the scene by distance or image distribution. The sub-networks are trained separately and composed with manually defined rules. The Switch-NeRF (MI & Xu, 2023) learns the scene decomposition by an MoE network and trains different experts in an end-to-end manner. There are also several methods (Xu et al., 2023; zha, 2023) employing the hash encoding (Müller et al., 2022) and plane encoding (Chan et al., 2022; Chen et al., 2022) while not decomposing the scene. In contrast to these existing works, our LeCO-NeRF method focuses on learning the occupancy of a large-scale scene. The learned occupancy can be used to accelerate large-scale NeRF methods.

**Occupancy and efficient sampling in NeRF.** Many methods are proposed to estimate the important regions. The original NeRF Mildenhall et al. (2020) trains a coarse and fine network together for hierarchical sampling. The Mip-NeRF 360 (Barron et al., 2022) designs a small proposal network to predict density and converts it into a sampling weight vector. Apart from these methods directly predicting the weight distributions, there are many methods (Müller et al., 2022; Fridovich-Keil et al., 2022; Hu et al., 2022; Li et al., 2022) use the binary occupancy for sampling. The Plenoxels method (Fridovich-Keil et al., 2022) reconstructs a sparse voxel grid and prunes empty voxels during the training. The NerfAcc (Li et al., 2022) provided a plug-and-play occupancy grid module and has shown in extensive experiments that estimating occupancy can greatly accelerate the training of various NeRF methods. The Instant-NGP (Müller et al., 2022) uses multi-scale occupancy grids to encode the occupancy. These existing methods using occupancy grids typically focus on small-scale scene modeling. The occupancy grid faces problems on large-scale scenes, as described above. In this paper, we focus on learning a compact binary occupancy representation on large-scale scenes.

**Mixture of experts.** The representative Mixture-of-Experts method (Shazeer et al., 2017) proposes Sparsely Gated Mixture of Experts (MoE). It selects different experts for different inputs by a gating network. The MoE has been applied to build large-scale models for various fields, such as NLP (Lepikhin et al., 2021; Fedus et al., 2022) and Computer Vision (Riquelme et al., 2021; Hwang et al., 2022). The Switch-NeRF (MI & Xu, 2023) is the first to successfully apply MoE on large-scale NeRF. Our method also uses MoE in NeRF. However, Our model has fundamental differences from Switch-NeRF: **(i)** The gating network in Switch-NeRF is homogeneous and not controllable. In contrast, our model is the first to control the MoE to learn the imbalanced occupancy of NeRF in a self-supervised manner. **(ii)** Learning occupancy with MoE relies on our proposed novel designs of the Heterogeneous MoE, the imbalanced gate loss, and the density loss.

## 3 METHOD

### 3.1 OVERVIEW

Our LeCO-NeRF learns occupancy of a 3D scene in the training of a Neural Radiance Field $F$. The proposed framework of LeCO-NeRF is shown in Fig. 2. $F$ takes a 3D point $\mathbf{x}$ and its direction $\mathbf{d}$ as input. It predicts the color $\mathbf{c}$ and density $\sigma$ for each $\mathbf{x}$. Each 3D point is processed independently. To effectively learn the scene occupancy, we design a Heterogeneous Mixture-of-Experts (HMoE) network structure. It contains an occupancy network $O$, several expert networks, and prediction heads. The occupancy network is a typical gating network in MoE structure. We have two parts of expert networks: the scene experts and an empty space expert. The scene experts consist of a set of $n$ scene expert networks $\mathcal{E}_s = \{E_i, i = 1...n\}$, and the empty space expert is designed to be a special tiny network $E_e$. The occupancy is encoded in the gate network $O$ in HMoE. The scene experts share a prediction head $H_s$. The empty space expert has its own prediction head $H_e$.

An input 3D point $\mathbf{x}$ of LeCO-NeRF first goes through the occupancy network $O$ and obtains $n + 1$ gate values. These values correspond to the $n$ scene experts and the empty space expert. Then, $\mathbf{x}$ will be dispatched into only one expert according to the gate values. If a scene expert is selected, this indicates that this $\mathbf{x}$ is occupied. $\mathbf{x}$ will be input to this expert and then goes through the prediction

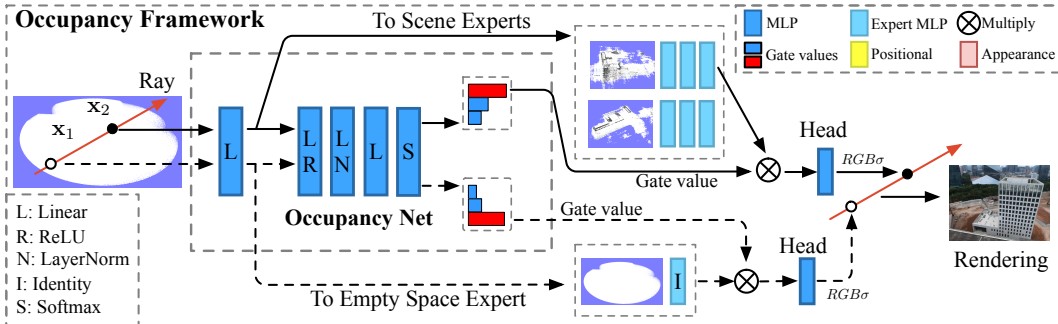

Figure 2: Our proposed LeCO-NeRF. The occupancy network serves as a gating network of our HMoE. $\mathbf{x}_1$ and $\mathbf{x}_2$ go through the occupancy network and are dispatched to the Empty Space Expert and a scene expert respectively according to the gate values. The occupancy can be trained together end-to-end with the NeRF network by multiplying the gate values on the output of the experts. If a point is dispatched into the Empty Space Expert, it is classified as unoccupied. The occupancy network is a small MLP. We enlarge the figure of the occupancy network to clearly show its operation.

head $H_s$ for $\sigma$ and $\mathbf{c}$. If the empty expert is selected, this represents that this $\mathbf{x}$ is not occupied by the scene. It will go through $E_e$ and $H_e$ to directly predict $\sigma$ and $\mathbf{c}$. After processing all the sampled points along a ray $\mathbf{r}$, the pixel color $C(\mathbf{r})$ is accumulated by volume rendering, and we compute a rendering loss between the rendered pixel color and the corresponding ground truth pixel color.

## 3.2 HETEROGENEOUS MIXTURE OF EXPERTS (HMoE)

**Experts and heads.** The proposed HMoE is heterogeneous because the experts and their operations are different. The scene experts $\mathcal{E}_s = \{E_i, i = 1...n\}$ contain $n$ experts with the same architecture. They are used to encode the points occupied by the scene. In our implementation, each scene expert contains 7 linear layers. The prediction head $H_s$ for $\mathcal{E}_s$ are shared. $H_s$ also accepts the view direction $\mathbf{d}$ and appearance embedding AE (Martin-Brualla et al., 2021) as inputs to encode a view-dependent color. The Empty Space Expert $E_e$ is defined as a tiny network. It is used to encode unoccupied (*i.e.* empty space) points. We use an identity layer in our implementation to directly feed forward the input. $H_e$ is the prediction head for $E_e$, which only takes the output of $H_e$ for predictions. The tiny $E_e$ results in fewer parameters for empty space. As a result, the empty space network tends to predict smooth values and therefore favors the empty space whose density is small and smooth. The scene experts $\mathcal{E}_s$ contain much more network parameters than the empty space expert $E_e$, because the occupied points have much more important information to be modeled.

**Occupancy network as a gating network.** The occupancy network $O$ in our HMoE serves as a gating network to dispatch input 3D points to different experts. The architecture of the occupancy network is shown in Fig. 2. $O$ predicts a vector of $n + 1$ normalized gate values $O(\mathbf{x})$ for a input 3D point $\mathbf{x}$. The first $n$ gate values correspond to the $n$ scene experts. The last gate value corresponds to the empty space $E_e$. We apply a Top-1 operation to $O(\mathbf{x})$ and obtain the index $k$ of the Top-1 value. Then, we dispatch $\mathbf{x}$ into the expert of index $k$. The gate value is multiplied by the output of the expert. This enables the occupancy network to be trained together with the whole network architecture. If $\mathbf{x}$ is assigned to $E_e$, it means $\mathbf{x}$ is unoccupied. It goes through $E_e$ and $H_e$ to predict $\sigma$ and color $\mathbf{c}$. If the assigned expert is one of the scene experts, this indicates $\mathbf{x}$ is an occupied surface point. Then $\mathbf{x}$ goes through the corresponding scene expert and is input into the head $H_s$ to predict $\sigma$ and $\mathbf{c}$. After training the whole network, the occupancy of a 3D scene can be encoded into the designed compact occupancy network. we can use the occupancy network $O$ as an occupancy predictor. An input point is unoccupied if the occupancy network dispatches it to $E_e$. In our implementation, the occupancy network contains 4 linear layers and a layer-norm layer.

## 3.3 OCCUPANCY OPTIMIZATION LOSSES

Based on the designed Heterogeous Mixture-of-Experts (HMoE) structure, we further propose an imbalanced gate loss and a density loss to regularize the optimization of the occupancy network and improve the accuracy of the occupancy, as shown in Fig. 3(a).

**Imbalanced gate loss.** Common MoE methods typically use a loss to ensure each expert obtains roughly the same number of input points. This gate loss helps the different experts to be fully

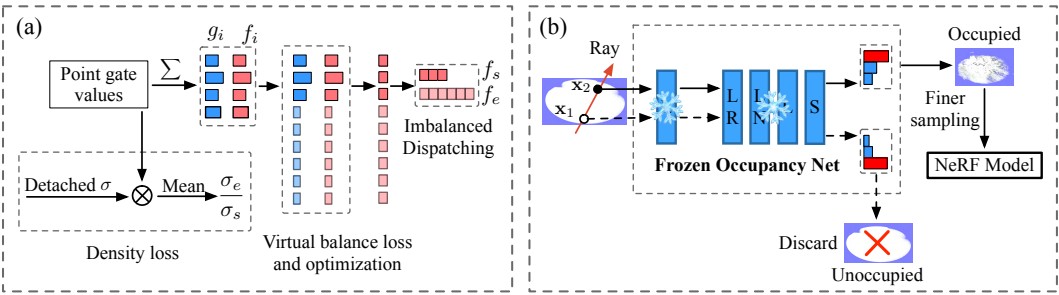

Figure 3: (a) the computation of imbalanced gate loss and density loss from gate values. (b) After training the occupancy network of a scene, we can use our frozen occupancy network to guide the sampling and training of NeRF methods.

utilized. However, in our framework, the empty space expert should naturally secure more 3D points, because a large portion of the 3D scene points is unoccupied. If we dispatch a similar number of points to the scene experts and the empty space expert, the learned occupancy is not sparse enough, and it does conform to the essential sparsity nature of the scene. Using it to guide the 3D point sampling does not help to largely reduce the number of scene points to be processed, thus no benefits for speeding up the training. Therefore, we design an imbalanced gate loss $L_g$, which can not only dispatch more points into $E_e$, but also can keep the number of points roughly the same for each scene expert. This means that $L_g$ is imbalanced for the empty space expert while balanced for the scene experts. We first introduce the balanced gate loss $L_b$ described in (Lepikhin et al., 2021). Let $n$ be the number of total experts, and $f_i$ be the fraction of points dispatched into expert $i$. Then $\sum f_i^2$ is minimized if all $f_i$ are equal. However, $\sum f_i^2$ is not differentiable, so it cannot be used as a loss function. As shown in (Lepikhin et al., 2021), we can replace one $f_i$ by a soft version $p_i$, where $p_i$ is the fraction of the gate values dispatched to expert $i$. Therefore, the balanced gate loss can be defined as $L_b = n \sum_{i=1}^{n} f_i p_i$. Under the optimal balance gating, $L_b$ will be 1. Inspired by $L_b$, we define the imbalanced loss $L_g$. We can consider $E_e$ as $v$ virtual experts. The fraction of each virtual expert is thus $f_e/v$, and the fraction of the gate values is $p_e/v$. Then, we can compute the balanced loss for the $v$ virtual experts and the $n$ scene experts. When $n+v$ experts are balanced, $E_e$ can obtain more points. Therefore, we define $L_g$ by modifying $L_b$ as follows:

$$L_g = (n + v) \left( v \frac{f_e}{v} \frac{p_e}{v} + \sum_{i=1}^{n} f_i p_i \right) = (n + v) \left( \frac{f_e p_e}{v} + \sum_{i=1}^{n} f_i p_i \right) \tag{1}$$

When optimal dispatching is achieved, $E_e$ obtains a portion of $v/(n + v)$ points. Each scene expert obtains a portion of $1/(n + v)$. $L_g$ is 1.0. This definition of $L_g$ is beneficial for setting loss weights because its scale is invariant with respect to $v$. This makes it convenient to set the loss weight for $L_g$. We set $n = 8$, $v = 80$ in our experiments. These values typically make the occupancy network dispatch about 85% points to the empty space expert.

**Density loss.** We design a density loss to explicitly guide the occupancy network to learn better occupancy. Our main idea is that the average density of the points dispatched to the empty space expert $E_e$ should be much smaller than that of the scene experts $\mathcal{E}_s$. Let the set of points dispatched to $E_e$ and $\mathcal{E}_s$ be $\mathcal{X}$ and $\mathcal{Y}$, respectively. The average density $\sigma_e$ of $E_e$ is $\sigma_e = \frac{1}{|\mathcal{X}|} \sum_{i \in \mathcal{X}} \sigma_i$. The average density $\sigma_s$ for $\mathcal{E}_s$ is $\sigma_s = \frac{1}{|\mathcal{Y}|} \sum_{i \in \mathcal{Y}} \sigma_i$. Then, the ratio $\sigma_e/\sigma_s$ should be small if the occupancy is learned correctly. The problem is that the $\sigma_e/\sigma_s$ cannot affect the gating network. Similar to the balanced gate loss, we include the gate values in the computation of the mean density. The $\sigma_e$ and $\sigma_s$ can be rewritten as $\sigma_e = \frac{1}{|\mathcal{X}|} \sum_{i \in \mathcal{X}} g_i \sigma_i$ and $\sigma_s = \frac{1}{|\mathcal{Y}|} \sum_{i \in \mathcal{Y}} g_i \sigma_i$. The value $g_i$ used for a point of the scene experts is the sum of the gate values for the $n$ scene experts. Therefore, the density loss $L_d$ can be defined as:

$$L_d = \frac{\sigma_e}{\sigma_s} = \frac{|\mathcal{Y}|}{|\mathcal{X}|} \frac{\sum_{i \in \mathcal{X}} g_i \sigma_i}{\sum_{i \in \mathcal{Y}} g_i \sigma_i} \tag{2}$$

We detach $\sigma$ when computing $L_d$, which is similar to the distillation training strategy utilized in several NeRF methods (Srinivasan et al., 2021; Barron et al., 2022). When $L_d$ is large, it optimizes the output of the occupancy network to make it dispatch correctly.

**Rendering loss.** Our network learns the occupancy during the training of NeRF. Therefore, our main optimization loss is the rendering loss (Mildenhall et al., 2020). We sample $N$ 3D points along a ray $\mathbf{r}$ and predict the density $\sigma_i$ and color $\mathbf{c}_i$ for each 3D point $\mathbf{x}_i$ by the network. We use $\sigma_i$ to compute $\alpha_i = 1 - \exp(-\sigma_i \delta_i)$, where $\delta_i$ is the distance of two nearby points. Then we compute the transmittance $T_i = \exp(-\sum_{j=1}^{i-1} \sigma_j \delta_j)$ of $\mathbf{x}_i$ along the ray. The predicted color $\hat{C}(\mathbf{r})$ is computed as $\hat{C}(\mathbf{r}) = \sum_{i=1}^{N} T_i \alpha_i \mathbf{c}_i$. The rendering loss $L_r$ is computed by $\hat{C}(\mathbf{r})$ and the ground-truth color $C(\mathbf{r})$. Let the set of rays be $\mathcal{R}$. $L_r$ is defined as $L_r = \sum_{r \in \mathcal{R}} \left\| \hat{C}(\mathbf{r}) - C(\mathbf{r}) \right\|_2^2$.

**Final loss.** The final optimization loss $L_f$ of our method is the weighted sum of $L_r$, $L_g$ and $L_d$. $L_f = w_r L_r + w_g L_g + w_d L_d$, where $w_r$, $w_g$, and $w_d$ are their corresponding loss weights.

### 3.4 OCCUPANCY AS GUIDANCE

After the occupancy network of a large-scale 3D scene is trained, we can freeze it and use it to guide the 3D point sampling of NeRF, shown in Fig. 3(b). We first sample a set of coarse samples and input them into the occupancy network $O$ and filter empty space points. Then we split the reserved samples to get finer sampling. This can reduce the number of points requiring evaluating the occupancy network. The unoccupied empty space points are discarded, typically 85% points from our observation. Since $O$ is much smaller than the main NeRF network, the training can be significantly accelerated. In the experiments, we typically sample 128 samples along each ray and use the occupancy to filter the sample and split each occupied sample into 8 new samples.

## 4 EXPERIMENTS

### 4.1 DATASETS

We use two publicly available large-scale datasets for evaluation. The Mega-NeRF dataset is adapted by Mega-NeRF (Turki et al., 2022) from its Mill 19 dataset and the UrbanScene3D (Liu et al., 2021), consisting of the Building, Rubble, Residence, Sci-Art, and Campus scenes. Each of them contains from $2k$ to $6k$ images with a resolution of about $5k \times 3k$. The Block-NeRF dataset Tancik et al. (2022) contains a scene with $12k$ images with a resolution of about $1k \times 1k$.

### 4.2 METRICS AND VISUALIZATION

We evaluate the occupancy accuracy with Occupancy Metrics and apply the occupancy on the sampling of Switch-NeRF (MI & Xu, 2023) to compute the Image Reconstruction Metrics.

**Occupancy metrics.** We evaluate the classification accuracy of the occupancy. The ground-truth occupancy is usually not available in real-world large-scale NeRF datasets. Since a fully-trained NeRF without using occupancy can get a good estimation of the geometry of the scene, we use it as a good reference for evaluation. We extract depth maps predicted by vanilla Switch-NeRF (MI & Xu, 2023) and convert them into an occupancy grid. Then we also convert our

Table 1: Accuracy, Precision, Recall and F1-Score, parameter number and occupancy ratio of different occupancy methods. Our method clearly outperforms the occupancy grid with a compact parameter size and occupancy ratio.

| Dataset | Method | Accu. | Preci. | Recall | F1 | Para. | Occ. |
|---|---|---|---|---|---|---|---|
| Sci-Art | Grid | **0.912** | 0.315 | 0.519 | 0.392 | 2.0M | 22.8% |
|  | Ours | 0.904 | **0.339** | **0.795** | **0.476** | **0.15M** | **13.0%** |
| Campus | Grid | 0.619 | 0.371 | 0.746 | 0.496 | 2.0M | 34.0% |
|  | Ours | **0.684** | **0.437** | **0.883** | **0.584** | **0.15M** | **14.5%** |
| Rubble | Grid | 0.697 | 0.269 | 0.666 | 0.383 | 2.0M | 33.6% |
|  | Ours | **0.712** | **0.319** | **0.914** | **0.473** | **0.15M** | **13.0%** |
| Building | Grid | 0.521 | 0.183 | 0.549 | 0.274 | 2.0M | 44.0% |
|  | Ours | **0.711** | **0.322** | **0.683** | **0.438** | **0.15M** | **15.0%** |
| Residence | Grid | 0.656 | 0.232 | 0.634 | 0.339 | 2.0M | 37.5% |
|  | Ours | **0.703** | **0.314** | **0.958** | **0.473** | **0.15M** | **15.9%** |

learned occupancy into another occupancy grid by sampling and evaluating point occupancy. The occupancy accuracy is computed by comparing the converted occupancy grids.

**Image reconstruction metrics.** We apply our learned occupancy on a state-of-the-art large-scale Switch-NeRF (MI & Xu, 2023) method. We use PSNR, SSIM (Wang et al., 2004) (both higher is better) and LPIPS (Zhang et al., 2018) (lower is better) to evaluate the validation images.

**Occupancy visualization.** We visualize the occupancy as point clouds. We sample and merge 3D points of rays in the validation images. These point clouds are visualized by two methods. The first one is to directly visualize the predicated color of each point. The second one uses the $\alpha = 1 - \exp(-\sigma_i \delta_i)$ as an additional channel to show the color and transparency of the point clouds. The unoccupied points should be largely transparent. The two visualization methods complement each other for better visualization of the occupancy.

### 4.3 IMPLEMENT DETAILS

For the training of occupancy of our LeCO-NeRF on Mega-NeRF dataset, we use 8 scene experts and 1 empty space expert. The gating network contains 1 input layer, 2 inner layers, 1 layernorm and 1 output layer. The channel of the number of the main layers is set as 256. we set $w_r = 1.0$, $w_g = 0.0005$, $w_d = 0.1$ and $v = 80$. We sample 512 points for each ray. We train the occupancy for $40k$ steps. The training of the occupancy network takes from 1.6h to 1.8h. The learning rate is set as $5 \times 10^{-4}$. For the training of occupancy of our LeCO-NeRF on Block-NeRF dataset, we use Mip embedding proposed in Barron et al. (2021). The channel of the number of the main layers is set as 512. $w_d$ is set as 0.005. $v$ is set as 40.

To apply our learned occupancy on Switch-NeRF (MI & Xu, 2023), we replace their sampling in their foreground NeRF with the guided sampling by the learned occupancy. To compare with the occupancy grid, we employ the OccGridEstimator from NeRFAcc Li et al. (2022) in Swtich-NeRF. The grid size is set as default $128^3$. The OccGridEstimator is updated along with the main network. Other training settings and network structures remain the same as the original Switch-NeRF. The main results

Table 2: The image accuracy on large-scale Block-NeRF dataset (Tancik et al., 2022). Our method not only outperforms Switch-NeRF (MI & Xu, 2023), but also outperforms the occupancy grid method by a PSNR of 0.84.

| Method | PSNR↑ | SSIM↑ | LPIPS↓ | Time (h)↓ |
|---|---|---|---|---|
| Switch-NeRF | 22.85 | 0.742 | 0.515 | 23.8h |
| Switch+Grid | 22.26 | 0.740 | 0.511 | 23.8h |
| Switch+Ours | **23.10** | **0.751** | **0.498** | 23.8h |

are trained on 8 NVIDIA RTX 3090 GPUs. We sample 1024 rays for each GPU for Mega-NeRF dataset and 1664 rays for Block-NeRF dataset. We align the training time based on the grid method.

### 4.4 BENCHMARK PERFORMANCE

**Occupancy Metrics.** We evaluate our occupancy accuracy with the Occupancy Metrics. Since the unoccupied and occupied points are highly imbalanced, we report the Accuracy, Precision, Recall and F1-Score to complement each other. As shown in Table 1, our learned occupancy can clearly outperform the Occupancy Grid in almost all the metrics in all Mega-NeRF dataset, with compact parameter sizes. Notably, our network is much better on Recall, indicating that it is good at correctly predicting the occupied point, which is critical for better NeRF optimization. The occupancy ratio in Table 1 means the ratio of points retained to go through the main NeRF. Our occupancy network also retains fewer points than the occupancy grid while getting better accuracy. This means our network can predict more accurate and compact occupancy.

**Image Reconstruction Metrics.** We report the PSNR, SSIM, LPIPS of applying our learned occupancy on Switch-NeRF (MI & Xu, 2023) on Block-NeRF in Table 2. Our method can outperform the occupancy grid by a large margin and outperform Switch-NeRF with the same training time. Note that the training time of our method includes our occupancy training time for fair comparison.

Table 3: The accuracy, training time and memory on large-scale Mega-NeRF dataset (Turki et al., 2022). Our method (S+Ours) clearly outperforms Switch-NeRF (S-NeRF*) (MI & Xu, 2023) and the occupancy grid (S+Grid) methods. Our method also consumes less memory when being used to guide the training of Switch-NeRF. S-NeRF* is trained with the same time as S+Grid.

| Dataset | Metrics | M-NeRF | S-NeRF | S-NeRF* | S+Grid | S+Ours |
|---|---|---|---|---|---|---|
| Sci-Art | PSNR↑ | 25.60 | 26.52 | 25.46 | 25.48 | **26.04** |
| | SSIM↑ | 0.77 | 0.795 | 0.762 | 0.760 | **0.772** |
| | LPIPS↓ | 0.390 | 0.360 | 0.400 | 0.413 | **0.398** |
| | Time | 31.7h | 45.4h | 14.1h | 14.1h | 14.1h |
| | Mem. | 5.6G | 10.5G | 10.5G | 5.8G | **2.7G** |
| Campus | PSNR↑ | 23.42 | 23.62 | 22.76 | 22.75 | **23.21** |
| | SSIM↑ | 0.537 | 0.541 | 0.507 | 0.500 | **0.517** |
| | LPIPS↓ | 0.618 | 0.609 | 0.659 | 0.671 | **0.635** |
| | Time | 32.0h | 42.4h | 12.5h | 12.5h | 12.5h |
| | Mem. | 4.8G | 10.1G | 10.1G | 6.0G | **2.3G** |
| Rubble | PSNR↑ | 24.06 | 24.31 | 23.58 | 23.69 | **23.96** |
| | SSIM↑ | 0.553 | 0.562 | 0.519 | 0.522 | **0.548** |
| | LPIPS↓ | 0.516 | 0.496 | 0.546 | 0.549 | **0.516** |
| | Time | 29.5h | 41.5h | 13.5h | 13.5h | 13.5h |
| | Mem. | 5.0G | 10.3G | 10.3G | 6.1G | **2.4G** |
| Building | PSNR↑ | 20.93 | 21.54 | 20.50 | 20.33 | **20.64** |
| | SSIM↑ | 0.547 | 0.579 | 0.517 | 0.495 | **0.522** |
| | LPIPS↓ | 0.504 | 0.474 | 0.526 | 0.547 | **0.517** |
| | Time | 30.7h | 42.5h | 13.7h | 13.7h | 13.7h |
| | Mem. | 5.0G | 10.2G | 10.2G | 6.1G | **2.4G** |
| Residence | PSNR↑ | 22.08 | 22.57 | 21.77 | **22.18** | 22.10 |
| | SSIM↑ | 0.628 | 0.654 | 0.611 | 0.622 | **0.626** |
| | LPIPS↓ | 0.489 | 0.457 | 0.501 | 0.500 | **0.485** |
| | Time | 32.9h | 43.4h | 14.8h | 14.8h | 14.8h |
| | Mem. | 5.4G | 10.5G | 10.5G | 6.4G | **2.9G** |

The results on Mega-NeRF dataset are in Table 3. The results of Mega-NeRF (M-NeRF) (Turki et al., 2022) and Switch-NeRF (S-NeRF) (MI & Xu, 2023) are taken from their papers. The S+Ours mean using our learned occupancy for sampling in Switch-NeRF. The S+Grid means using an occupancy grid for sampling. S-NeRF* are the results

of vanilla Switch-NeRF trained with the same time as S+Grid for fair comparison. We highlight the best values among S-NeRF*, S+Grid and S+Ours. Note that we include the occupancy training time in S+Ours for a fair comparison. As shown in Table 3, when trained with the same time, our method consistently outperforms vanilla Switch-NeRF and the occupancy grid. Therefore, our method is significant to speed up the training of Switch-NeRF while getting competitive accuracy. We visualize the point clouds of occupancy in Fig. 7 in Appendix A. The point clouds show that our network can learn more compact and clean occupancy than the occupancy grid. We also provide the visualization comparison of rendered images in Appendix B and a video in the supplementary files.

### 4.5 ABLATION STUDY

In this section, we perform several ablations to analyze the designs of our HMoE, the density loss, and the learned occupancy. The experiments are performed by applying our occupancy on Switch-NeRF and the Sci-Art scene for 40K occupancy steps and 500K NeRF steps if not specified.

**HMoE.** We perform experiments to show our Heterogeneous Mixture of Experts (HMoE) can implicitly learn the occupancy. We design a Homogeneous MoE by setting the empty space expert the same large as the scene experts. As shown in Table 4, the rendering accuracy of the Homogeneous MoE (Homo.) largely dropped compared with our Heterogeneous Mixture of Experts (HMoE) (Heter.). Our HMoE can get reasonable accuracy. Note that the density loss is not used in these experiments. As shown in Fig. 4(a), the scene experts of HMoE handle the full occupied points. The scene experts of the Homogeneous MoE only handle a part of the occupied points. This means that the Homogeneous MoE cannot learn reasonable occupancy and its rendered image is thus of low quality. These experiments show that, to implicitly model the heterogeneous occupancy of a 3D scene, it is important to design a heterogeneous network structure.

Table 4: Ablation on Homogeneous MoE, Heterogeneous MoE with $L_g$ without the density loss $L_d$. Homogeneous MoE cannot learn reasonable occupancy while Heterogeneous MoE can learn good occupancy and get better accuracy.

| Method | PSNR↑ | SSIM↑ | LPIPS↓ |
|--------|-------|-------|--------|
| Homo. | 20.23 | 0.631 | 0.506 |
| Heter. | **26.30** | **0.781** | **0.383** |

**Density loss.** We ablate on the density loss $L_d$ on Building scene in Table 5. Our full method achieves better accuracy than that without density loss. As shown in Fig. 4(b), with $L_d$, the network can separate the occupied and unoccupied points more clearly and the rendered images contain fewer artifacts in the challenging region. These experiments show that our density loss can give more explicit information to the occupancy network and make the occupancy network learn more accurate occupancy.

Table 5: Ablation on the density loss $L_d$ on Building scene. $L_d$ can help our occupancy network learn better occupancy and get better accuracy.

| Method | PSNR↑ | SSIM↑ | LPIPS↓ |
|--------|-------|-------|--------|
| Ours w/o $l_d$ | 20.59 | 0.516 | 0.521 |
| Ours | **20.79** | **0.531** | **0.508** |

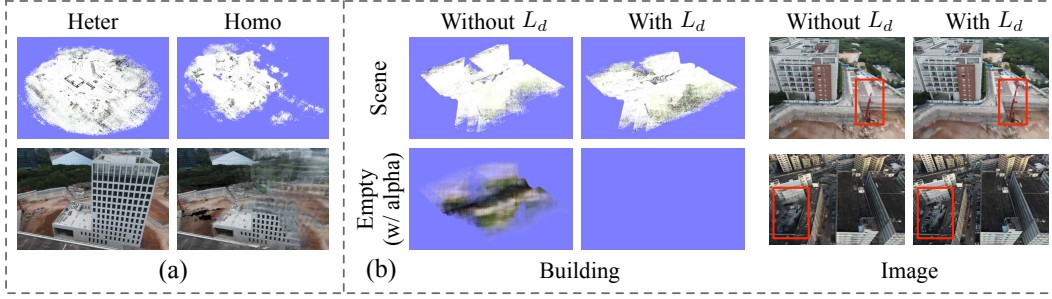

Figure 4: (a) The occupancy and images with Homogeneous MoE and our Heterogeneous MoE. The Heterogeneous MoE learns good occupancy and renders better images. The Homogeneous MoE cannot distinguish the occupied and unoccupied regions. (b) Point clouds of the scene experts and the empty space expert with and without the density loss. We visualize the point clouds of the empty space expert with transparency related to alpha values as described in Sec. 4.2 to better show whether the points are empty or not. $L_d$ can make our Heterogeneous MoE learn better occupancy thus the image for $E_e$ is all empty. With $L_d$, the images are more complete in challenging regions.

**Occupancy analysis.** We analyze the occupancy statistics related to the points of scene experts and the empty space expert with respect to the occupancy training steps. They are computed with the model of different training steps on the evaluation images of the Sci-Art scene. These values are shown in Fig. 5. We also visualize the point clouds of different training steps in Fig. 6 complementing the quantitative results.

Fig. 5a is the portion of points in scene experts $\mathcal{E}_s$ and the empty space expert $E_e$ of different training steps. There are consistently more than 80% points in $E_e$. The portion also increases during training. This figure proves the effectiveness of our imbalanced gate loss. It also means that we can speed up the training a lot if we use learned occupancy to guide the sampling of points. Fig. 5b and Fig. 5c show mean density values and alpha values of points in $\mathcal{E}_s$ and $E_e$. The points of $\mathcal{E}_s$ have clearly much larger densities and alpha values than those of $E_e$. The values of points in $E_e$ are nearly zero. This shows that our network can dispatch points according to their densities. Fig. 5d shows the density value ratio and alpha value ratio between points in $\mathcal{E}_s$ and $E_e$. The values of points in $\mathcal{E}_s$ are several magnitudes larger than those in $E_e$. The ratios are the direct target of the density loss so they validate the effectiveness of the design of the density loss.

Fig. 6 visualizes the point clouds dispatched to the scene experts and empty space experts. The points in the scene experts have larger opacity. They cover the whole scene surface quickly only after $10k$ steps of training. The points in the empty space expert consistently have very small opacity, indicating that they are empty. The first row visualizes points without transparency and the second row visualizes points with transparency.

The analyses of occupancy show clearly that LeCO-NeRF can learn the occupancy of a 3D scene accurately and fast. The occupancy can be effectively encoded into the compact occupancy network.

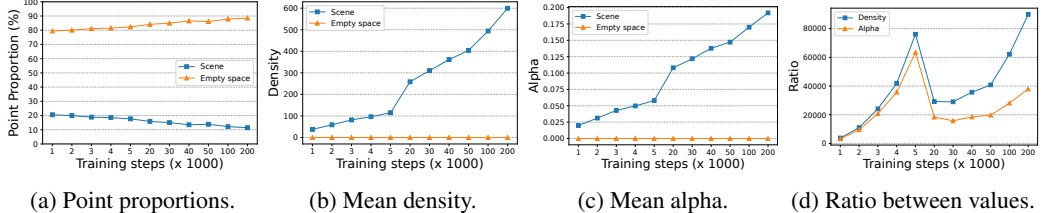

| (a) Point proportions. | (b) Mean density. | (c) Mean alpha. | (d) Ratio between values. |

Figure 5: The statistics of scene experts $\mathcal{E}_s$ and the empty space expert $E_e$ in different training steps. (a) The portion of points in $\mathcal{E}_s$ and $E_e$. (b) (c) The mean density values and alpha values of points in $\mathcal{E}_s$ and $E_e$. (d) The density value ratio and alpha value ratio between points in $\mathcal{E}_s$ and $E_e$.

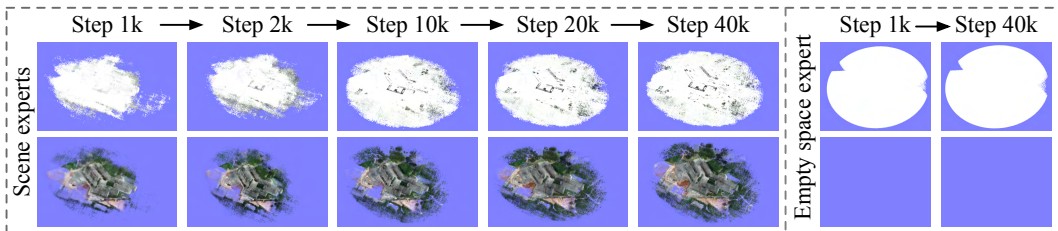

Figure 6: The point clouds dispatched to the scene experts and the empty space expert in each step. The scene experts converge fast to the whole occupied area. The points in the empty space experts consistently have very small opacity. Our network can learn accurate occupancy with only $20k$ to $40k$ steps. The two rows visualize the point clouds without and with transparency respectively as described in Sec. 4.2.

## 5 CONCLUSION

In this paper, we propose LeCO-NeRF to learn compact occupancy for large-scale scenes. We achieve by our core designs of a novel Heterogeneous Mixture of Experts (HMoE) structure, an imbalanced gate loss, and a density loss. Experiments on challenging large-scale datasets have shown that our learned occupancy clearly outperforms the occupancy grid and can achieve competitive accuracy with much less time. Since occupancy is a very important concept in many 3D research areas, this work will give more inspiration to the research of learning and representation of occupancy.

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

## APPENDIX

## A    OCCUPANCY POINT CLOUD

We compare the occupancy point cloud of our occupancy network and occupancy grid on Mega-NeRF dataset (Turki et al., 2022) in Fig. 7. The point clouds show that our network can learn more compact and clean occupancy than the occupancy grid.

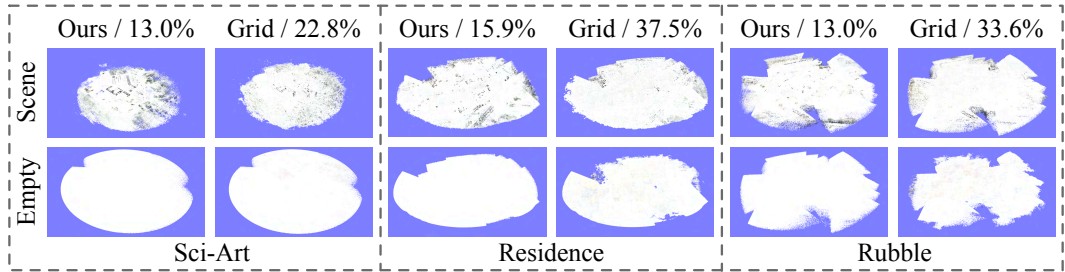

Figure 7: The visualization of our occupancy and grid occupancy as point clouds. Our predicted occupied points (scene) are cleaner and have fewer points than the occupancy grid. They fit the surface of the buildings more compactly.

## B    RENDERED IMAGES

We compare the rendered images of our occupancy network and occupancy grid on Mega-NeRF dataset (Turki et al., 2022).

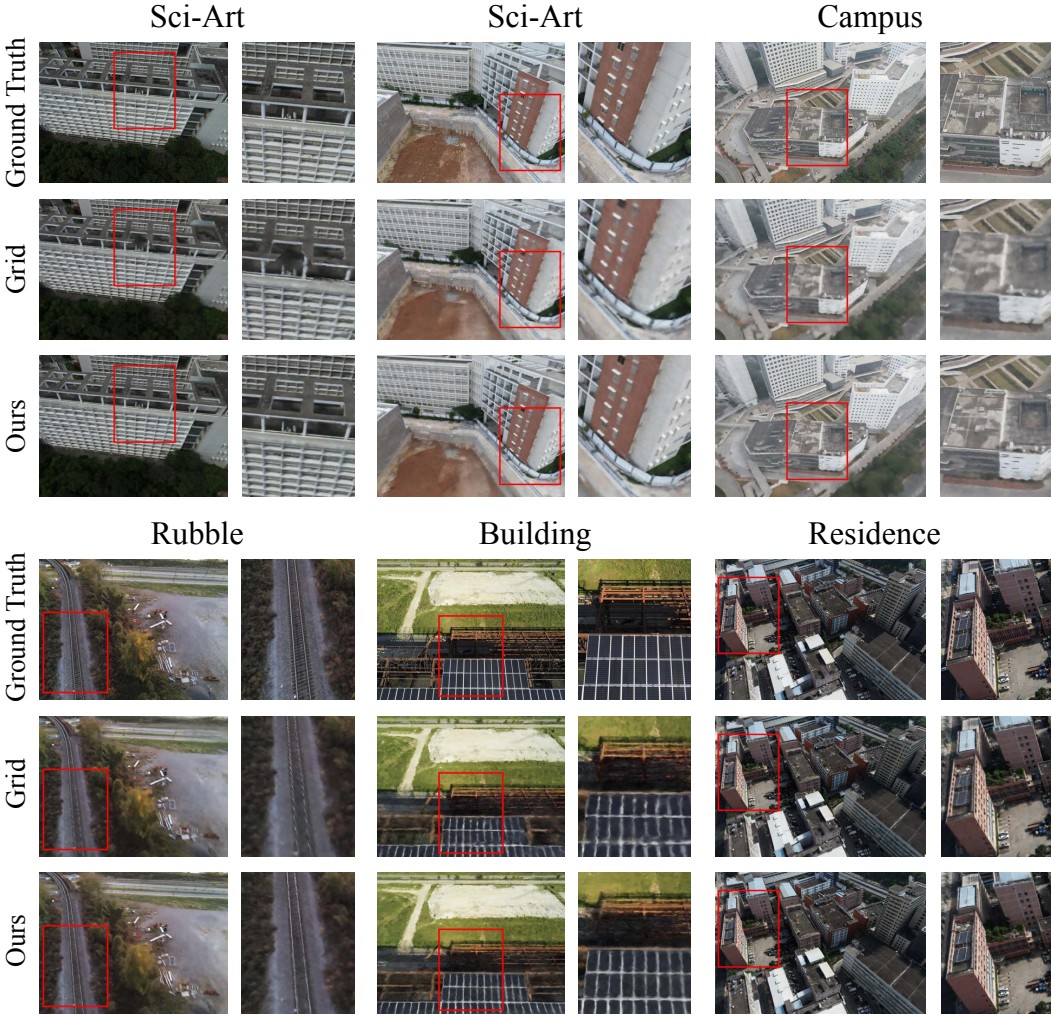

Figure 8: The rendered images of our occupancy network and the occupancy grid. Our method can get more complete, clean, and high-quality images than those of the occupancy grid.

## C    INSTANT-NGP AS BASELINE

Apart from the Switch-NeRF (MI & Xu, 2023), we use our learned occupancy network to guide the 3D point sampling of a fast NeRF method Instant-NGP (Müller et al., 2022). The Instant-NGP uses efficient hash encodings for NeRF representation and the occupancy grid for empty space skipping. In our experiments on Instant-NGP, we adapt the Instan-NGP into the unbounded large-scale dataset. For the foreground, we use the standard hash grids in Instant-NGP. For the background, we contract the space into a bounded space and define hash grids on the contracted space, following the contraction in Mip-NeRF 360 (Barron et al., 2022) and Nerfacc (Li et al., 2022). An occupancy grid is used to guided the sampling in Instant-NGP. We train the Instant-NGP for 500K steps on 2 GPUs with a batch size of 8192. We also use our learned occupancy network in our main experiments to guided the training of the hash encodings (Ours+NGP). We align the training time of (Ours+NGP) to the traing time of Instant-NGP. A shown in Table 6, our method (NGP+Ours) clearly outperform

the original Instant-NGP by large margins. Since the original Instant-NGP use the occupancy grid to cache the occupancy, the results clearly demonstrate that our learned compact occupancy network can learn much better occupancy than the occupancy grid.

Table 6: The accuracy, training time on large-scale Mega-NeRF dataset (Turki et al., 2022) with Instant-NGP as baseline. When training with the same time, our method (NGP+Ours) clearly outperforms the original Instant-NGP (Müller et al., 2022). by large margins. Since the original Instan-NGP use the occupancy grid to cache the occupancy, the results clearly show that our compact occupancy network can learn much better occupancy than the occupancy grid.

| Dataset | Metrics | NGP | NGP+Ours |
|---------|---------|------|----------|
| Sci-Art | PSNR↑ | 23.98 | **24.50** |
| | SSIM↑ | 0.724 | **0.754** |
| | LPIPS↓ | 0.445 | **0.412** |
| | Time | 12.9h | 12.9h |
| Campus | PSNR↑ | 21.76 | **22.83** |
| | SSIM↑ | 0.475 | **0.518** |
| | LPIPS↓ | 0.677 | **0.623** |
| | Time | 12.7h | 12.7h |
| Rubble | PSNR↑ | 22.94 | **23.66** |
| | SSIM↑ | 0.498 | **0.558** |
| | LPIPS↓ | 0.572 | **0.501** |
| | Time | 10.4h | 10.4h |
| Building | PSNR↑ | 19.48 | **20.33** |
| | SSIM↑ | 0.454 | **0.511** |
| | LPIPS↓ | 0.585 | **0.518** |
| | Time | 12.42 | 12.42 |
| Residence | PSNR↑ | 21.27 | **21.77** |
| | SSIM↑ | 0.591 | **0.626** |
| | LPIPS↓ | 0.515 | **0.473** |
| | Time | 11.0h | 11.0h |

# D ACCURACY OF DIFFERENT TRAINING TIMES.

We analyze the detailed accuracy of our method on Sci-Art with respect to the training time in Fig. 9. Our method (S+Ours) demonstrates remarkable convergence speed compared to the grid-based occupancy (S+Grid) and the original Switch-NeRF (MI & Xu, 2023) (S-NeRF). Note that the training time of our method in this figure includes training time of our occupancy network.

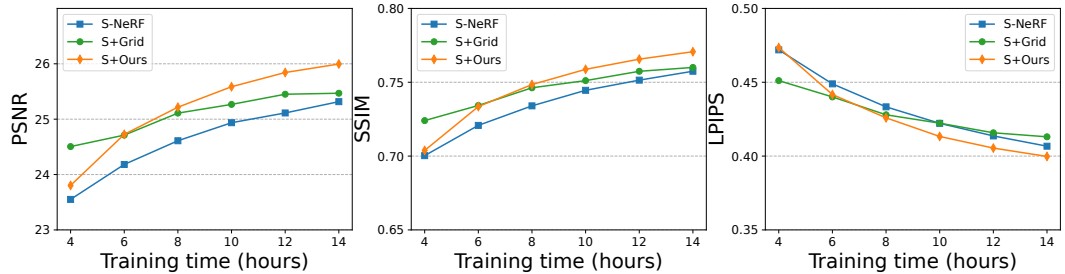

Figure 9: Analysis of training time and accuracy: Our method (S+Ours) demonstrates remarkable convergence speed when compared to grid-based occupancy (S+Grid) and the original Switch-NeRF (MI & Xu, 2023) on Sci-Art. Note that the training time for our method includes the training time of our occupancy network.

# E    ABLATION ON GRID RESOLUTION.

We analyze the effect of the resolution of the occupancy grid to better show the advantage of our compact occupancy network. In the main experiments, we use an occupancy grid of $128^3$. In this ablation study, we use an occupancy grid of $256^3$ to guide the training of Switch-NeRF. As shown in Table 7, the occupancy grid with a resolution of 256 (Grid-256) consumes dramatically more time and memory for training. It takes over 46 hours for training of 500K steps, which is much slower than Grid-128, and still gets worse results than ours trained by 14.1h. Moreover, Grid-256 consumes about $5\times$ memory than our method. This study shows clearly the advantage of the compactness of our occupancy network.

Table 7: Ablation study on the resolution of the occupancy grid. With a resolution of 256 (Grid-256), the occupancy grid method takes over 46 hours for training of 500K steps, which is much slower than Grid 128, and still gets worse results than ours trained by 14.1h. Moreover, Grid-256 consumes about $5\times$ memory than our method.

| Method | PSNR↑ | SSIM↑ | LPIPS↓ | Time↓ | Mem.↓ |
|---|---|---|---|---|---|
| Grid-128 | 25.48 | 0.760 | 0.413 | 14.1h | 5.8G |
| Grid-256 | 25.86 | 0.764 | 0.409 | 46.0h | 12.5G |
| Ours | **26.04** | **0.772** | **0.398** | 14.1h | **2.7G** |

# F    TRAINING TIME

In the main experiments, we train the grid-based method for $500k$ steps and report the results of our method aligning with the training time of the grid based method. In this ablation study, we align the training with the original Switch-NeRF to analyze the accuracy of our method with respect to the training time. As shown in Table 8

Table 8: The ablation of the same more training time

| Dataset | Metrics | S-NeRF | S+Grid | S+Ours |
|---|---|---|---|---|
| Rubble | PSNR | 24.31 | 23.94 | **24.52** |
| | SSIM | 0.562 | 0.538 | **0.581** |
| | LPIPS | 0.496 | 0.526 | **0.477** |
| | Time | 41.5h | 41.5h | 41.5h |
| Residence | PSNR | 22.57 | 22.48 | **22.88** |
| | SSIM | 0.654 | 0.645 | **0.658** |
| | LPIPS | 0.457 | 0.474 | **0.454** |
| | Time | 43.4h | 43.4h | 43.4h |

## G ABLATION ON SIZE OF EMPTY SPACE EXPERT.

We perform ablation study on the size of our proposed expert space expert in Table 9 below. In the table, Homo. is a large MLP with 7 layers. The 4-layer version can learn better occupancy wile the Identity version can gets results very close to Switch-NeRF. Therefore, from our extensive experiments, with an Identity layer as the expert space expert can learn very good results. The output of the Identity empty space expert will still go through the prediction head so the network can still be trained.

Table 9: Ablation on the size of the expert space expert.

| Method | PSNR↑ | SSIM↑ | LPIPS↓ |
|--------|-------|-------|--------|
| Homo | 20.23 | 0.631 | 0.506 |
| 4-layer | 23.62 | 0.701 | 0.463 |
| Identity | **26.30** | **0.781** | **0.383** |

