# OpenReview forum: "LeCO-NeRF: Learning Compact Occupancy for Large-scale Neural Radiance Fields"
_ICLR.cc/2024/Conference — Submitted to ICLR 2024_

### Official Review · Reviewer_iRL2 · 2023-10-22

**Soundness:** 2 fair
**Presentation:** 2 fair
**Contribution:** 2 fair
**Rating:** 5
**Confidence:** 4

**Summary:**

This work aims to learn and encode the occupancy of a scene into a compact MLP in an efficient and self-supervised manner.
First, this paper proposes a Heterogeneous Mixture of Experts (HMoE) structure with common Scene Experts and a tiny Empty-Space Expert. Second, an imbalanced gate loss is proposed for HMoE, motivated by the prior that most of the 3D points are unoccupied. It enables the gating network of HMoE to accurately dispatch the unoccupied and occupied points. Third, an explicit density loss is introduced to guide the gating network. Then, the occupancy of the entire large-scale scene can be encoded into a very compact gating network of the HMoE. With the learned occupancy as guidance for empty space skipping, our method can consistently obtain 2.5× speed-up on the state-of-the-art method Switch-NeRF, while achieving highly competitive performances on several challenging large-scale benchmarks.

**Strengths:**

This paper proposes a novel Heterogeneous Mixture of Experts (HMoE) network to learn the occupancy of a large-scale scene. The HMoE network consists of several Scene Experts designed to encode the occupied points.

An imbalanced gate loss is introduced for the gating network in HMoE. Since a large portion of the space is unoccupied, the decision of the gating network can explicitly model the imbalance of occupancy.

To better learn the occupancy of a large-scale scene, a density loss is proposed to guide the training of the gating network.

Experiments show that after training, the occupancy network can be utilized to guide the point sampling in the large-scale NeRF method, i.e. Switch-NeRF (MI & Xu, 2023), and achieve a 2.5× acceleration compared to Switch-NeRF, while obtaining highly competitive performance.

**Weaknesses:**

This paper devotes large efforts to learning and encoding the occupancy of a large-scale scene into a compact MLP, which can be used to guide the point sampling (mainly for skipping the points in empty space) of other NeRF models.
However, I have several major concerns about this work.

There are many representations to cache the occupancy/density of the scene, such as the MLP, grid, hashgrid, tensoRF, and triplanes. This paper only compares a grid representation with a resolution of 128^3. Even under this setup, the improvement of S+Ours over S+Grid is minor and even worse in PSNR on Residence (S+Grid 22.18 vs S+Ours 22.10), as shown in Table 3.

I think using a larger resolution (e.g., 256) of the grid can achieve better results. I understand that using a large resolution, dense grid will occupy more memory. In this case, this method should compare with the more efficient hashgrid representation, which can represent a very high-resolution grid (e.g., 1024) with small memory and be optimized in a small amount of time (typically a few minutes on a single GPU). In comparison, the proposed occupancy method requires 1.6h to 1.8h to train on 8 NVIDIA RTX 3090 GPUs.

Overall, I feel that this method is heavy and the advantages over existing representations are very limited.

**Questions:**

Please kindly provide comparisons with more efficient representations, such as hashgrid, TensoRF, and K-planes. And justify why we need such a complicated method to learn occupancy.

Please also compare the runtime of different methods in learning the occupancy.

---

> ### Author Response · Authors · 2023-11-23
> **Response to Reviewer iRL2**
>
> We thank the reviewer for the valuable and insightful comments. We address the detailed questions and comments below.
>
> **Q1: Differences between the representations of the binary occupancy and NeRF.** Many thanks for this question. We would like to clarify the fundamental differences between the representations of the binary occupancy and Neural Radiance Fields. The representations of MLP, hashgrid and tensoRF can only be used to represent the Neural Radiance Fields in previous methods. They are not used for the representation of the binary occupancy because their occupancy cannot be trained and thus cannot be encoded into the MLP, hashgrid and plane encodings. This is the reason that the KiloNeRF with MLP, the Instant-NGP with hashgrid, the TensoRF with plane features all use a dense grid for the binary occupancy. As far as we know, our method is the first to represent the binary occupancy by MLP and one of our contributions is to make the binary occupancy trainable by the MLP network. Therefore, for "I understand that using a large resolution, dense grid will occupy more memory. In this case, this method should compare with the more eîcient hashgrid representation,", it is not practical to represent the occupancy grid by a hash grid without our training strategies.
>
> For the hash representation of Neural Radiance Fields, we have added new experiments in Table 6 in Appendix C and the table in Reviewer KM5v Q4. In these experiments, our method outperform the Instant-NGP by large margins, i.e. 1.07 point of PSNR on Campus, 0.85 on Building, 0.72 on Rubble and over 0.5 on other scenes. These experiments clearly show that our compact occupancy network can learn much better occupancy than the occupancy grid on different baselines.
>
> We also add an ablation study on the resolution of the dense grid Table 7 in Appendix E and the table below. In this ablation study, we use an occupancy grid of $256^3$ to guide the training of Switch-NeRF. As shown in the table, the occupancy grid with a resolution of 256 (Grid-256) consumes dramatically more time and memory for training. It takes over 46 hours for training of 500K steps, which is much slower than Grid-128, and still gets worse results than ours trained by 14.1h. Moreover, Grid-256 consumes about $5\times$ memory than our method. This study shows clearly the advantage of the compactness of our occupancy network.
>
>
>
> | Method    | PSNR↑ | SSIM↑ | LPIPS↓ | Time↓ | Mem.↓ |
> | --------- | ----- | ----- | ------ | ----- | ----- |
> | Grid-128  | 25.48 | 0.760 | 0.413  | 14.1h | 5.8G  |
> | Grid-256  | 25.86 | 0.764 | 0.409  | 46.0h | 12.5G |
> | Ours      | **26.04** | **0.772** | **0.398**  | 14.1h | **2.7G**  |
>
> **Q2: Running time of learning the occupancy.** Many thanks for the question. As stated above, the representation of the 2D binary occupancy is not the same as the NeRF representations. The updating of the occupancy grid is along with the training of NeRF and is not separated.Therefore, we provide the running time of our method in the below table.
>
> | Scene     | Building | Rubble | Residence | Sci-Art | Campus |
> |-----------|----------|--------|-----------|---------|--------|
> | Time (h)  | 1.74     | 1.73   | 1.80       | 1.76    | 1.65   |

---

### Official Review · Reviewer_uh8i · 2023-10-31

**Soundness:** 3 good
**Presentation:** 3 good
**Contribution:** 3 good
**Rating:** 5
**Confidence:** 5

**Summary:**

The proposed LECO-Nerf is a reconstruction method for large scale scenes. The major contribution is the Occupancy Net that predicts whether a 3D point is occupied or not. If occupied, the Occupancy Net predicts which expert (i.e., nerf network) the point belongs to. To facilitate the training of the Occupancy Network, the Imbalanced Gate Loss is used to allocate more points to the empty expert, the Density Loss is used to enforce that points classified as empty are of small occupancy \sigma. Experiments show that the training speed is improved, with competitive reconstruction quality.

**Strengths:**

1. It is the first method that models the occupancy space with a MLP. Experiments show that the design is effective. Most points are empty, and the predicted sigmas are close to zero for those empty points.
2. The imbalanced Gate Loss is inspiring to control the number of points belongs to each nerf network.

**Weaknesses:**

1. The HMoE design is questionable. According to Section 4.5 HMoE, the reconstruction quality droped with MoE (The empty expert is as large as scene experts).
(a) I believe it is the density loss that enforces the small sigma for empty points. If a larger empty expert leads to failure, that means the density loss is invalid / not-effective.
(b) If a larger empty expert fails, the authors need to justify how "small" the empty expert should be. It would be impractical if the HMoE is sensitive to the network design. Probably it requires different network designs when working with different scenes.

2. I am confused about the statement "we can use our frozen occupancy network to guide the sampling and training of NeRF methods". The occupancy network can be trained end-to-end according to the design. Why do we need to freeze the occupany network and train the nerf models again?

3. Unfair experiment design. Most experiments are of fixed training time, e.g. 23.8h for Table 2, 12-15h for Table 3. Indeed the HMoE gives better results with limited training time. However, we need to compare the final result with enough training time. Does the HMoE still out-performs the baselines with enough training time, e.g., 40+ hours for Table 2 & 3.

4. For Figure 2, why do we need an empty space expert? Why do we need sigma and RGB if we believe a point is empty? An obvious alternative is that occupancy network serves as a classifier. That is:
(a) Remove the empty space expert
(b) The empty points do not involve in the rendering, in another word, set \sigma=0 if a point is predicted as empty.

**Questions:**

Overall I like the idea of modeling occupancy with a MLP. I would lean to acceptance if the above questions are solved.

---

> ### Author Response · Authors · 2023-11-23
> **Response to Reviewer uh8i (part 1/2)**
>
> We thank the reviewer for the valuable and insightful comments, and the positive comments such as "the first method", "effective" and "I like the idea of modeling occupancy with a MLP". We address the detailed questions and comments below.
>
> **Q1: Homogeneous MoE and Heterogeneous MoE.** We are very sorry for the misleading caption of Table 4 and the text in the Section 4.5 HMoE. We forget to mention the experiment details of this ablation study. In Table 4, both the Homo. and Heter. MoE are trained with the imbalanced gate loss and without density loss for 40K steps and than their occupancy networks are used to guide the training of Switch-NeRF. The accuracy in Paper Table 4 are the final reconstruction accuracy of the guided stage. These experiments are to show that with our HMoE and imbalanced gate loss, our method can already learn very good occupancy. The Homogeneous MoE with imbalanced gate loss **cannot** learn reasonable occupancy so if we use its occupancy network to guide the training of Switch-NeRF, many important (occupied) points will be classified as unoccupied. Therefore, the accuracy drops a lot. We have fixed this misleading components in the revised version. In our main experiments, the empty space expert is set as an Identity layer and have been shown as effective across quite different large-scale scenes. We also add experiments of different sizes of the empty space experts in Table 9 in Appendix G and the table below. In the table, Homo. is a large MLP with 7 layers. The 4-layer version can learn better occupancy wile the Identity version can gets results very close to Switch-NeRF. Therefore, from our extensive experiments, with an Identity layer as the expert space expert can learn very good results. The output of the Identity empty space expert will still go through the prediction head so the network can still be trained.
>
>
> | Method    | PSNR↑ | SSIM↑ | LPIPS↓ |
> |-----------|-------|-------|--------|
> | Homo      | 20.23 | 0.631 | 0.506  |
> | 4-layer   | 23.62 | 0.701 | 0.463  |
> | Identity  | **26.30** |**0.781** | **0.383**  |
>
> **Q2: Guided training.** The main motivation of freezing the occupancy network is to decrease the computation in the training of NeRF and to sample denser samples. In the training of occupancy network, we uniformly samples 3D points along rays. And the unoccupied points still need gradient and optimization for training the occupancy which consumes more memory and computation. Since our occupancy network converges very fast, we can freeze the occupancy network to accelerate the classification of 3D points and sampling denser samples to train NeRF methods, without extra gradients and optimization for the occupancy network.

---

> > ### Author Response · Authors · 2023-11-23
> > **Response to Reviewer uh8i (part 2/2)**
> >
> > **Q3: Training time in experiment design.** In the main experiments, we fix the same training time to clearly show the advantage of our method to get better results with the same training time. As stated in Section 4.3, the training times are aligned to the training time of S+Grid for 500k steps. We revised this part to make it more clear. We add a curve figure to show the accuracy with respect to the training time in Fig. 9 in Appendix D, as suggested by Reviewer KM5v Q3. As shown in this figure, our method enjoys better accuracy abd fast converging speed. We also add Table 8 in Appendix F to train S+Grid and S+Ours with the same time as the original Switch-NeRF, as shown in the table below and Table 8, HMoE can consitently out-performs the baselines with enough training time.
> >
> >
> >
> > | Dataset    | Metrics | S-NeRF | S+Grid | S+Ours |
> > |------------|---------|--------|--------|--------|
> > | Rubble     | PSNR    | 24.31  | 23.94  | **24.52** |
> > |            | SSIM    | 0.562  | 0.538  | **0.581** |
> > |            | LPIPS   | 0.496  | 0.526  | **0.477** |
> > |            | Time    | 41.5h  | 41.5h  | 41.5h  |
> > | Residence  | PSNR    | 22.57  | 22.48  | **22.88** |
> > |            | SSIM    | 0.654  | 0.645  | **0.658** |
> > |            | LPIPS   | 0.457  | 0.474  | **0.454** |
> > |            | Time    | 43.4h  | 43.4h  | 43.4h  |
> >
> >
> > **Q4: The importance of empty space expert.** The empty space expert is a key component to make the occupancy network trainable. The occupancy network cannot distinguish the points before the training. It will get false classification in the beginning of the training. We need the empty space expert and its prediction to backpropagate the gradients and give training signals to the occupancy network. Therefore, we need to predict the sigmas and RGBs of points in the empty space expert. The rendering loss, imbalanced gating loss and density loss all need the output of the empty space expert to train the occupancy network in a self-supervised manner. As for "An obvious alternative is that occupancy network serves as a classifier. That is: (a) Remove the empty space expert (b) The empty points do not involve in the rendering, in another word, set sigma=0 if a point is predicted as empty.", it is actually our guided stage in the paper using our frozen occupancy network to guide the training of NeRF methods. In contrast, in the stage of training occupancy network, we need the empty space expert to pass training signals.

---

### Official Review · Reviewer_Zume · 2023-11-01

**Soundness:** 3 good
**Presentation:** 4 excellent
**Contribution:** 3 good
**Rating:** 8
**Confidence:** 2

**Summary:**

The paper introduced LeCO-NERF, a novel representation of large scenes. There are several novelties of the paper, including a new Heterogeneous mixture of expert structure to effectively model occupied/unoccupied regions in NERF, as well as corresponding loss functions to guide the learning of this new network. As a result, the learned NERF is highly accurate and compact with the help of the occupancy network. The proposed algorithm on public benchmarks achieved 2.5x speed-up while maintaining state-of-the-art accuracy.

**Strengths:**

- The paper is trying to solve a very important problem of learning a compact and efficient occupancy representation of large-scale scenes.
- The introduction of the Heterogeneous Mixture of Experts (HMoE) structure is novel, and ablation studies shows this design is essential
- The newly introduced loss functions are proven to be useful as well in practice
- LeCO-NERF is highly efficient compared to state of the art approaches, while getting very high accuracy on public benchmarks on large scenes
- The overall presentation of the paper is good

**Weaknesses:**

I think this is a nice paper, and I couldn't find highly concerning issues and weaknesses. I do have some minor questions, and they will be listed in the following section

**Questions:**

- The paper introduced an algorithm to handle very large scenes. However I wonder if there are intuitions that can be shared on 1) how big of the scene can the algorithm handle in theory (and in practice), and 2) how small of the scene can LeCO-NERF become effective (against existing SOTA NERF variants for 3D objects).
- I spent considerable amount of time trying to understand the blue box with white contents of Fig.1, before learning more context about what it represents. I wonder if this visualization can be better improved by providing more explicit notations in the caption or in the main text
- in Fig 4b, with L_d,  it seems that the rendering is all blank, is this a mistake?
- will the code be available upon publication? I think this would be useful for reproducing this research work

---

> ### Author Response · Authors · 2023-11-23
> **Response to Reviewer Zume**
>
> We thank the reviewer for the valuable and insightful comments, and the positive comments such as "solve a very important problem", "novel" and "useful". We address the detailed questions and comments below.
>
> **Q1: Intuitions of the scene scales.** Many thanks for this question. We describe the large-scale scenes in our experiments to give more intuitions. The Mega-NeRF dataset comprises 5 scenes each consisting of $2k$ to $6k$ images with a resolution from $4k$ to $5k$, covering urban regions from 0.15 to 1.3 $km^2$. The Block-NeRF dataset includes a street scene with $12k$ images at $1k \times 1k$ resolution, covering an Urban street of $1km$ long. These scenes are very challenging due to the huge image data and the coverage of large-scale regions. The experiments have shown that our compact occupancy network can handle these large-scale scenes effectively. From our perspective, for the small scale objects, the efficiency problem is not as critical as the large-scale scene because of the efficient representation such as the hash-encoding, small data volume, dense multi-view images and requirements of small resolution of grids. In the large-scale urban scenes, due to the large data volume and large scene range, the learning of occupancy becomes much harder and critical, and our proposed compact occupancy network can effectively solve these problems. Our newly add experiments on large-scale scnens with Instant-NGP as baseline also show that it can handle different large-scale representations (Appendix C, Table 6 and Reviewer KM5v Q4).
>
>
> **Q2: Visualization in Fig. 1.** Many thanks for this suggestion. We add more details about the visualization in the caption of Fig. 1 and the main text.
>
> **Q3: Blank image in Fig 4b.** Many thanks for the question. This is not a mistake. The bottom right image of Fig 4b shows the unoccupied points with alpha. Points with small alpha will be shown as transparent. Please refer Section 4.2 Occupancy visualization for more details. Since with density loss, our method can predict more accurate occupancy, the points in this image are all with very small density and alpha. Therefore, all of the points of the bottom right image of Fig 4b are transparent when shown with alpha and thus the image is all blank. Compare with the bottom left image, the bottom right image shows the effectiveness of our density loss to separate the occupied and unoccupied points more clearly. We add more description of this image in the caption and main text in the revised version.
>
> **Q4: Code releasing.** We will surely release our codes upon publication to benefit the community.

---

### Official Review · Reviewer_KM5v · 2023-11-06

**Soundness:** 3 good
**Presentation:** 3 good
**Contribution:** 2 fair
**Rating:** 5
**Confidence:** 4

**Summary:**

This paper proposes a new approach to learning compact occupancy for large-scale neural radiance fields (NeRF) called LECO-NERF. The authors address the challenge of efficiently encoding the occupancy of a scene into a compact MLP, which can be used to guide empty-space skipping and point sampling. They propose a Heterogeneous Mixture of Experts (HMoE) structure to model unoccupied and occupied regions in NeRF, which is trained using a novel imbalanced gate loss. The authors demonstrate that LECO-NERF achieves state-of-the-art results on several large-scale datasets while being more efficient and compact than previous methods. Overall, the paper presents a promising approach to improving the scalability and efficiency of NeRF for modeling large-scale scenes.

**Strengths:**

Overall the paper is based on a similar idea of Switch-NeRF, and developed based on that by adding novel modules including HMoE and imbalanced gate loss. Strengths including:
- The proposed Heterogeneous Mixture of Experts (HMoE) structure is a novel way to model unoccupied and occupied regions in NeRF and is trained using a novel imbalanced gate loss.
- The authors demonstrate that LECO-NERF achieves state-of-the-art results on several large-scale datasets while being more efficient and compact than previous methods.
- The paper provides a detailed analysis of the occupancy statistics related to the points of scene experts and the empty space expert with respect to the occupancy training steps, which helps to understand the behavior of the model.

**Weaknesses:**

My major concern with the paper is limited novelty and improvement over existing literature on large-scale neural rendering, as well as inadequate evaluation of related benchmarks.

- The general idea of using MoE to improve the performance of large-scale neural radiance fields has been explored by Switch-NeRF. The work is rather incremental by modifying the homogenous mixture of experts into the heterogeneous mixture of experts (HMoE).
- The reconstruction quality, when compared to the original switch-nerf, seems to be only on par or marginally improved (Tab. 2, Tab.3)
- Tab.3 is confusing, can you elaborate more on how S-NeRF* is different from S-NeRF? Maybe include a curve showing how the test PSNRs change against training time.
- The reviewed and evaluated baselines in this paper are quite limited. Sparse occupancy-based methods (\eg NSVF, PointNeRF, 3D Gaussian Splatting, nerflets) are not covered in related works. Possible hash-encoding, plane-encoding, or hybrid-encoding methods for large-scale NeRFs are not covered adequately as baselines in the main experiments.

**Questions:**

Check the weaknesses for details:

- Improve the related work section.
- Adding/discussing more possible baselines in the main experiments.
- Discuss more about how this method improves over the plain MoE/Switch-NeRF.
- Including some visualizations (of reconstructed images) for the main experiment.
- Including more details about Tab. 3.
- Minor: Tab. 3 caption: ->MegaNeRF.

---

> ### Author Response · Authors · 2023-11-23
> **Response to Reviewer KM5v (part 1/2)**
>
> We thank the reviewer for the valuable and insightful comments. We address the detailed questions and comments below.
>
> **Q1: The difference between our method and Switch-NeRF.** We would like to clarify our motivation, novelty, and our distinctive features compared to Switch-NeRF.
>
> In this paper, we solve a fundamentally different problem compared to Switch-NeRF in large-scale NeRF modeling. The Switch-NeRF aims to learn the decomposition of a large-scale scene for scalable training. Our paper targets learning a compact representation of a binary occupancy of a large-scale scene, which can provide effective guidance to learn a large-scale NeRF efficiently. This is the first time that this problem has been introduced for large-scale NeRF modeling.
>
> To tackle this important and new problem of modeling the occupancy of a large-scale scene, our approach is designed based on our insights on the problem, and our contribution is three-fold:
>
> **(i) A novel heterogeneous MoE structure with scene experts and a tiny Empty Space Expert.** This structure is designed to effectively model the high heterogeneity of occupied and unoccupied 3D scene points.
>
> **(ii) A novel and effective gate imbalanced loss.** If we train the HMoE with the balanced loss as used in the original Switch-NeRF that targets a homogeneous modeling of the scene, the model cannot learn the occupancy because the scene distribution is highly heterogeneous with a lot more points belonging to the empty space compared to the surface. It is thus very important to enforce a training prior to make the network distinguish the occupied and the unoccupied 3D points. The proposed gate imbalanced loss can effectively control the portion of points dispatched to the empty space expert, which is a critical design for the successful large-scale occupancy learning.
>
> **(iii) A novel and effective density loss.** Our designed density loss can directly control the gating network according to the predicted density. It enables our occupancy network to explicitly model the important scene geometry prior, which is the density of an unoccupied scene point is much smaller than that of an occupied scene point. The density loss helps to ensure the learning of more accurate scene occupancy.
>
> In the experiments of the main paper, our method has shown consistent advantages of accuracy, converging speed, and the GPU memory overhead over the occupancy grid method and the Switch-NeRF trained with the same time. In our newly added experiments (Table 6 in the appendix and in Q4) with the Instant-NGP as a baseline, our method also outperforms the Instant-NGP by large margins. These experiments show that our method gains consistent performance improvements compared to the occupancy grid method in various datasets and baselines, demonstrating the effectiveness of our proposed approach for learning a compact large-scale scene occupancy.
>
> In conclusion, our paper tackles a fundamentally different problem compared to Switch-NeRF and all our three novel contributions are closely related to our insights on the new problem. Our method confirms consistent advantages over the occupancy grid method.
>
> **Q2: Reconstruction quality.** We would like to highlight our clear advantages of accuracy, converging speed, and the GPU memory overhead over the occupancy grid method and the Switch-NeRF, which have been extensively verified in our experimental results. For example, our model achieves an improvement of about 0.6 PSNR over Switch-NeRF on Sci-Art (Table 3 in the main paper) when being trained with the same time. We obtain a similar PSNR with only 8 hours training compared to Switch-NeRF using 14 hours (Fig. 9 in the main paper). Our method also uses much less memory (2.4G) than Switch-NeRF (10.5G) and the occupancy grid (5.8G) as reported in Table 3 of the main paper. In our newly added experiments (Table 6 in the appendix and also in Q4 of the rebuttal) with the Instant-NGP as the baseline, our method also outperforms the Instant-NGP by large margins, e.g., 1.07 points of PSNR on Campus and 0.85 on Building. These experiments on the different datasets and baselines demonstrate the effectiveness and the advantages of our proposed approach.

---

> > ### Author Response · Authors · 2023-11-23
> > **Response to Reviewer KM5v (part 2/2)**
> >
> > **Q3: S-NeRF\* and S-NeRF in Table 3 and a new figure of PSNRs changing with time.** We are sorry for the confusion. S-NeRF\* and S-NeRF are trained with different times. The S-NeRF denotes the Switch-NeRF trained with 500k steps. The results are obtained from the original paper of Switch-NeRF. The S-NeRF* is the Switch-NeRF trained with the same time the Switch-NeRF+Grid (S+Grid). We align the training time to S+Grid for a fair comparison. We have revised the caption of Table 3 to make it more clear.
> >
> > Thanks for the suggestion of a figure of showing PSNRs changing against the training time. We add Fig. 9 in Appendix D. It shows that our approach (S+Ours) obtains clearly better accuracy and faster converging speed.
> >
> >
> > **Q4: More related works and hash-encoding as baseline.** Thanks for the suggestion of adding more related works and baselines to better validate the contribution of our method. We revised the paper to cover more works in the related work section. We added new experiments in Appendix C and Table 6 by adding new baselines as well. In these experiments, the Instant-NGP is an adapted version for the large-scale NeRF modeling. It contains hash encodings for both the foreground and background. It uses an occupancy grid for the empty space skipping. We further use our learned occupancy network to guide the sampling of the Instant-NGP. As shown in the table below and Table 6 in the main paper, our method  (NGP+Ours) demonstrates superior performance compared to the Instant-NGP (NGP). Notably, our method outperforms the Instant-NGP by large margins, i.e. 1.07 points of PSNR on Campus, 0.85 on Building, 0.72 on Rubble, and over 0.5 on other scenes. These experiments clearly show that our proposed compact occupancy network can learn much better occupancy than the occupancy grid.
> >
> >
> > | Dataset    | Metrics    | NGP   | NGP+Ours |
> > |------------|------------|-------|----------|
> > | Sci-Art    | PSNR↑      | 23.98 | **24.50** |
> > |            | SSIM↑      | 0.724 | **0.754** |
> > |            | LPIPS↓     | 0.445 | **0.412** |
> > |            | Time       | 12.9h | 12.9h    |
> > | Campus     | PSNR↑      | 21.76 | **22.83** |
> > |            | SSIM↑      | 0.475 | **0.518** |
> > |            | LPIPS↓     | 0.677 | **0.623** |
> > |            | Time       | 12.7h | 12.7h    |
> > | Rubble     | PSNR↑      | 22.94 | **23.66** |
> > |            | SSIM↑      | 0.498 | **0.558** |
> > |            | LPIPS↓     | 0.572 | **0.501** |
> > |            | Time       | 10.4h | 10.4h    |
> > | Building   | PSNR↑      | 19.48 | **20.33** |
> > |            | SSIM↑      | 0.454 | **0.511** |
> > |            | LPIPS↓     | 0.585 | **0.518** |
> > |            | Time       | 12.42 | 12.42    |
> > | Residence  | PSNR↑      | 21.27 | **21.77** |
> > |            | SSIM↑      | 0.591 | **0.626** |
> > |            | LPIPS↓     | 0.515 | **0.473** |
> > |            | Time       | 11.0h | 11.0h    |
> >
> > **Q5: Visualizations of reconstructed images.** Thanks for the suggestion. We provide Figure 8 in Appendix B for the visualization of the reconstructed images. We also provide a video demo in the supplementary files.
> >
> > **Q6: Revision of paper.** Thanks for the suggestions about the paper revision. We added more related papers in the related work section, and added new experiments related to the hash-encoding baseline in Appendix C and Table 6 show the improvements of our method over baselines. We fixed the typo and added more details to Table 3.

---

### Author Response · Authors · 2023-11-23
**To all reviewers**

**Paper changes.**
1. Add related works and revised captions and texts according to the suggestion of reviewers.
2. Add the experiments with Instant-NGP as baseline in Appendix C, Table 6.
3. Add analysis of accuracy with different training time in Appendix D and Fig. 9.
4. Add ablation of the grid resolution of the occupancy grid in Appendix E and Table 7.
5. Add accuracy of more training time in Appendix F and Table 8.
6. Add ablation on size of our proposed empty space expert in Appendix G and Table 9.


We appreciate a lot if you kindly check the new experiments and new figures supporting the novelty and advantage of our method.

---

### Meta-Review · Area_Chair_Pvp7 · 2023-12-13

**Metareview:**

This paper receives 3x marginally below the acceptance threshold and 1x accept, good paper. The major weaknesses pointed out by the reviewers are: limited novelty and improvement over existing literature on large-scale neural rendering, as well as inadequate evaluation of related benchmarks. The general idea of using MoE to improve the performance of large-scale neural radiance fields has been explored by Switch-NeRF. Results also show marginal improvement over switch-nerf. The HMoE design is questionable. Unfair experiment design. There are many representations to cache the occupancy/density of the scene, such as the MLP, grid, hashgrid, tensoRF, and triplanes. This paper only compares a grid representation with a resolution of 128^3.

**Justification For Why Not Higher Score:**

The novelty of the proposed method is questionable as compared to swtichNeRF and the experimental results are also unfair.

**Justification For Why Not Lower Score:**

N/A

---

### Decision · Program_Chairs · 2024-01-16

Reject